# Tsunami Hazard Assessment in the South China Sea Based on Geodetic Locking of the Manila Subduction Zone

Guangsheng Zhao[1], Xiaojing Niu[1]

[1] State Key Laboratory of Hydroscience and Engineering, Department of Hydraulic Engineering, Tsinghua University, Beijing, 100084, China.

*Correspondence to*: Xiaojing Niu (nxj@tsinghua.edu.cn)

**Abstract.** This study provides a dataset and shows the spatial distribution of tsunami hazard in the South China Sea sourcing from the Manila subduction zone. The plate motion data around the Manila subduction zone is used to invert the geodetic locking of the Manila subduction zone, further used to estimate the maximum possible magnitude, and applied to obtain a more reliable tsunami hazard assessment. The spatial distribution of tsunami wave height with 1000 years return period is shown and several high hazard areas in the South China Sea are pointed out. Uncertainties of the seismic source are explored, including the slip heterogeneity, the upper limit of seismic magnitude and segmentation. The impact of locking distribution and randomness of slip on tsunami hazard assessment demonstrates that traditional uniform slip assumption significantly underestimates the tsunami hazard. Moreover, the assessment results involving the effect of locking distribution should be more realistic, and show a larger tsunami height than only considering the stochastic slip in most areas, which prompt the coastal management agencies to enhance the tsunami prevention awareness.

**Short summary.** The purpose of this study is to estimate the spatial distribution of the tsunami hazard in the South China Sea from the Manila subduction zone. The plate motion data is used to invert the degree of locking on the fault plane. The degree of locking is used to estimate the maximum possible magnitude of earthquakes and describe the slip distribution. A spatial distribution map of the 1000-year return period tsunami wave height in the South China Sea was obtained by tsunami hazard assessment.

## 1 Introduction

The South China Sea and surrounding coastal regions are at risk of earthquakes and tsunamis from the Manila Trench, which is the junction of the Philippine Sea plate and the Sunda plate. There is an absence of earthquakes with magnitudes greater than 7.6 since 1560 in the Manila subduction zone and the convergence rate is approximately 80mm/a, which means that an unresolved strain of up to approximately 38 m may have accumulated (Terry et al., 2017). Therefore, the Manila subduction zone is considered a potential source of tsunamis, which may result in megathrust rupture affecting the entire South China Sea. Tsunami deposits discovered about 1000 years ago in the Xisha Islands (Sun et al., 2013), Nan'ao Island coastal zone in

Guangdong Province (Yang et al., 2019), and Dongshan Bay in the northeast of the South China Sea (Huang et al., 2023) are consistent with the written records of catastrophic events characterized by giant waves in 1076, as evidence of a large earthquake in Manila subduction zone 1000 years ago and a destructive tsunami in the South China Sea. Different studies have estimated the maximum possible earthquakes in the Manila subduction zone, such as designing a magnitude 9.0 earthquake based on the geometric characteristics of the potential rupture zone (Megawati et al., 2009), or balancing the

accumulated strain in the Manila subduction zone with magnitude 8.8-9.2 earthquakes (Hsu et al., 2016), considering the worst-case scenario of the Manila subduction zone, assuming a magnitude of 9.35 (Wu and Huang, 2009) or 9.3 (Nguyen et al., 2014). However, in recent years, research on the north-south differences of the Manila subduction zone (Lin et al., 2015; Yu et al., 2018; Tan, 2020) seems to have weakened the possibility of large-scale megathrust earthquakes occurring in the Manila subduction zone. Therefore, the maximum possible earthquake in the Manila subduction zone remains a concern.

A great earthquake under the seabed may trigger a destructive tsunami to the coastal area. Accurately assessing the tsunami hazard is helpful for effective response measures. However, the tsunami hazard assessment is complex. On the one hand, many uncertain factors have an impact on the results of tsunami hazard assessment. In the entire process of tsunami hazard assessment, there are varying degrees of uncertainty in the physical characteristics of the source, the hydrodynamic characteristics of tsunami propagation and the inundation process (Behrens et al., 2021). Some traditional tsunami research

assumed that earthquake rupture was uniform. However, the seismic slip inversion of the 2004 Sumatra-Andaman tsunami (Lorito et al., 2010) and the 2011 Tohoku tsunami (Ozawa et al., 2011) showed significant spatial variability in the slip of large magnitude earthquakes. Therefore, the current description of earthquake rupture is generally based on the stochastic slip model. Due to the randomness of the earthquake source, a large number of tsunami scenarios are often required to quantify the tsunami threat through probabilistic tsunami hazard assessment. For example, Li et al. (2016) studied the impact

of uniform and heterogeneous slip distribution on the tsunami hazard assessment and the tsunami wave height with 1000-year return period of Hong Kong is about 2.0 m. Li et al. (2017) studied the role of upper magnitude limits in probabilistic tsunami hazard assessment and the tsunami hazard of Hong Kong at return period of 1000 years are about 0.5~3.5 m. Sepúlveda et al. (2019) conducted probabilistic tsunami hazard assessment focusing on the sensitivity to earthquake recurrence relationships, the maximum tsunami amplitude of 0.18 m is exceeded in Hong Kong with a mean return period of

100 years. Liu et al. (2021) considered the local and regional tsunami sources and the tsunami wave height of Hong Kong is 0.32 m for 475-year return period and 0.50 m for 975-year return period. Yuan et al. (2021) considered the tsunami source from both the South China Sea and the Northwest Pacific Ocean and the maximum wave amplitude of Hong Kong is about 2.5 m for 2000-year return period and 1.5m for 500-year return period. In some PTHA work, unit source or sub-fault methods are used to convert tsunami simulation into linear superposition of unit sources based on the linear characteristics of

tsunami waves in deep water, thereby reducing the computational complexity. And on the other hand, the largest source of uncertainty is the seismic activity including the maximum possible magnitude and rupture characteristics (Li et al., 2022). The maximum possible magnitude determines the upper limit of the magnitude for the tsunami scenarios in PTHA and has an impact on the magnitude frequency curve, thereby affecting the probability distribution of tsunami magnitude. The

maximum possible magnitude of an earthquake source is closely related to the cumulative rate of seismic moment, which

can be estimated through the geodetic locking model.

With the rapid development of Global Navigation Satellite System (GNSS), the method of using surface GPS horizontal velocity field data to invert fault locking and slip deficit has been widely used (Fletcher et al., 2001; Banerjee & Burgmann, 2002; Moreno et al., 2010; Ozawa et al., 2011; Ader et al., 2012; Li & Freymueller, 2018). So far, there already are some studies on locking inversion in the Manila subduction zone. Galgana et al. (2007) showed that the locking degree of the

Manila subduction zone is very low, with a locking coefficient of 0.01. Hsu et al. (2012) suggested that the Manila subduction zone is partially locked between 14.5-17.0°N, with an average locking coefficient of 0.4. Hsu et al. (2016) estimated that the locking coefficient of the Manila Trench at 15.0-19.0°N is 0.34~0.48. Those works are good references for the present study, but further analysis is still needed for deeply integrating the effect of locking distribution into PTHA. The fault locking can be used to evaluate the cumulative rate of seismic moment, thereby estimate the moment magnitude of

earthquake over a certain period of time. At the same time, studies have shown that in faults with higher degree of locking and accumulating slip deficit at a faster speed, the accumulated seismic moment of faults is greater and the likelihood of earthquakes is greater (Ozawa et al., 2011; Lamb et al., 2018). And when an earthquake occurs, the area with the highest slip may be related to asperities with high locking and slip deficit (Konca et al., 2008; Moreno et al., 2010; Moreno et al., 2011). Therefore, the distribution of locking and slip deficit can also be used to describe the characteristics of slip distribution in an

earthquake. However, the relationship between fault locking and slip in the next earthquake event is not absolute. The method of Small and Melgar (2021) is used to correct the probability of high slip in the high locking area.

The study aims to clarify the spatial distribution of tsunami hazard in the South China Sea through the PTHA considering geodetic locking. The effects of different heterogeneous slip distributions on PTHA results are investigated, as well as the impact of other source uncertainties such as the upper magnitude limits and source segmentation.

## 2 Methods

### 2.1 Inversion method

Fault locking models are obtained from inversion of GPS velocities by using inversion model TDEFNODE (McCaffrey 2009). TDEFNODE is used to model elastic lithospheric block rotations and internal strains, locking on block-bounding faults and so on. The model decomposes the relative displacement of blocks into the movement of the blocks on the

spherical surface and the deformation of the block boundary (McCaffrey, 2002). Generally, the locking distribution along the dip profile is assumed to have similarity, and parameterized functions are used as a primary guess of locking distribution. The Gaussian function and the Gamma function are widely used in the locking inversion. The Gaussian type refers to the distribution of locking coefficients along the dip profile as a Gaussian function; Gamma type refers to the exponential distribution of locking coefficients along the dip profile. The goal of inversion is to find the optimal parameters of the

assumed function that minimizes the chi-square value between the observed data and the model data.

The blocks involved include the Sunda plate, the Philippine Sea plate, and the Philippine mobile zone formed by the convergence of the Sunda plate and the Philippine Sea plate at the Philippines (Barrier et al., 1991; Aurelio, 2000). Referring to the previous study (Hsu et al., 2016), 9 blocks are divided around the Manila subduction zone. The Manila Trench subduction zone is located at the junction of the Sunda Plate and the Philippines Plate in the western part of Luzon Island.

The geometry of the Manila Trench subduction zone is interpolated using the Slab2 model (Hayes et al., 2018). In this study, only fault planes within a depth of 120 km are considered, and a total of 13 isobaths are set along the trend of the Manila Trench, with depths of 0, 10, 20, 30, 40, 50, 60, 70, 80, 90, 100, 110, and 120 km. Nodes are set at an interval of 0.5° on each bathymetric line, and there are 21 nodes on each bathymetric line.

The research selected GPS velocity field data (Kreemer et al., 2014), and converted to the ITRF08 reference framework. The

105 database contains observation data from about 80 GPS stations near the Manila Trench. In addition, about 50 GPS stations located in surrounding areas have been selected to limit the movement of the Sunda Plate and the Philippine Sea Plate. In the model, data with great uncertainty (velocity data uncertainty greater than 3.3 mm/a) were eliminated, and ultimately data of 144 GPS stations were used in the inversion.

## 2.2 Stochastic slip distribution with locking distribution as constraints

The Manila fault plane is divided into grids with a length and width of 0.1°, and there are 90×20=1800 sub faults in total. The longitude and latitude, depth, strike angle, dip angle, and rake angle of the center of each sub fault are interpolated from the data of Slab2. In each earthquake rupture event, the spatial distribution of rupture slip is projected onto the corresponding sub faults. On each sub faults, the slip is uniform, obtained by high-resolution slip generated from the stochastic slip model averaging.

Subsequently, the size of rupture needs to be determined. The size of rupture is calculated using the scaling relationship (Blaser et al., 2010). We did not use the same rupture size for the same magnitude. Instead, the variability of rupture length and width was preserved through random sampling.

In order to consider the locking distribution as constraints on heterogeneous slip on ruptures, the Code for Earthquake Rupture and ground-motion Simulation (CRES) (Mai & Beroza, 2002) is used in the study and the part of slip distribution

calculation is modified to consider geodetic locking model as constraints. In the CRES, the final slip distribution consists of two parts, one is the stochastic slip calculated by the spatial random field model mentioned above, and the other is the deterministic slip, which is uniformly distributed on the rupture, and the value is the average slip μ calculated based on the seismic scaling relationship. In order to consider the impact of geodetic locking model, we did not change the stochastic slip part, but scaled the deterministic part according to the locking distribution. Subsequently, the deterministic slip is

superimposed on the stochastic slip to obtain the final heterogeneous slip distribution. In the generated slip distribution, positions with high locking may not necessarily have high slip, but have a high probability of high slip. On average, the spatial distribution characteristics of slip is the same as the locking distribution.

## 2.3 Probabilistic tsunami hazard assessment

In probabilistic tsunami hazard assessment, the probability that the tsunami heights $H$ caused by all earthquake scenarios in a certain region within a certain period of time exceeds a critical value $h_c$ is:

$$p(H \geq h_c) = \sum_{i=1}^{N} p_i \cdot p(H \geq h_c \mid E_i) \tag{1}$$

where $N$ is the total number of possible earthquake scenarios. $p_i$ is the probability of earthquake event $E_i$ occurring. $p(H \geq h_c \mid E_i)$ is the probability that the tsunami height in the target area exceeds $h_c$ when an earthquake event $E_i$ occurs. $p_i$ is estimated statistically using the historical earthquake data. The historical earthquake data from 1900 to 2022 of the United States Geological Survey (USGS) is used to calculate the coefficients in the Gutenberg-Richter relationship.

The potential earthquake tsunami scenario set is a set that considers all earthquake magnitudes and epicenter positions. For each earthquake tsunami scenario, 100 stochastic slip distributions are considered. Due to the large number of earthquake tsunami scenarios, this study adopts the simplified and efficient solution (Miranda et al., 2014; Molinari et al., 2016; Zhang & Niu, 2020). Based on the linear assumption of tsunami waves in the deep-sea region, the simulation process of earthquake induced water surface disturbances and tsunami wave propagation is separated. The tsunami wave process is presented through the superposition of fluctuations generated by unit point source water level disturbances, and a large number of earthquake tsunami scenarios are analyzed using the propagation of water level disturbances of limited unit sources.

The study used the analytical model (Okada, 1985) to generate the initial disturbance of water surface during earthquakes. Unit sources with intervals of 0.1° are uniformly distributed in the region (116.5°E-123.5°E, 12°N-23°N), and the same Gaussian distribution initial water level is set on each point source. Gaussian distribution has the characteristics of symmetry and smoothness, which can effectively approximate complex initial water level fields. The initial water level distribution of unit sources will be stacked according to certain proportion coefficients to obtain the initial water level field of tsunami event calculated by the Okada model. The propagation of initial water surface disturbances of unit source is simulated using the ocean hydrodynamic model FVCOM.

It should be noted that due to the limitation of the superposition method, this study would not determine tsunami heights in coastal regions and the water depth of target points is all greater than 100 m. Tsunami waves propagating in shallow water will show complex non-linear behaviours, which can be simulated using non-linear models. The tsunami height in shallow water can be obtained approximately from the tsunami wave height at offshore point such as 100 m depth through multiplying the nearshore amplification factors (Glimsdal et al., 2019; Gao et al., 2022). Generally, the tsunami dataset in this study can be adopted as the boundary condition for detailed nearshore hazard analysis.

# 3 Results

## 3.1 Seismic potential of the Manila subduction zone

Fault locking is related to seismic potential in subduction zone. Generally, in higher locking patches, faults accumulate slip deficit at a faster rate, thereby accumulating more stress, and are more likely to become the start of rupture or generate
greater slip in the next earthquake event (Ozawa et al., 2011; Lamb et al., 2018). Fault locking and slip deficit are obtained from inversion of GPS velocities by using the inversion program TDEFNODE (McCaffrey, 2009). The observed GPS velocities distributed on the Philippine Sea Plate, the Sunda Plate and surrounding areas are used (Fig. 1a). Two widely-used locking distributions are considered in the inversion including the Gamma distribution and Gaussian distribution, which were both used to describe fault locking in relevant research (McCaffrey et al., 2013; Schmalzle et al., 2014). The fault locking
and slip deficit of two distribution are calculated (Fig. 1c and 1d). The chi-square value of Gamma and Gaussian distribution is 3.4 and 4.4, respectively, which indicates good inversion results. Limited data on fault locking cannot determine which distribution is better.

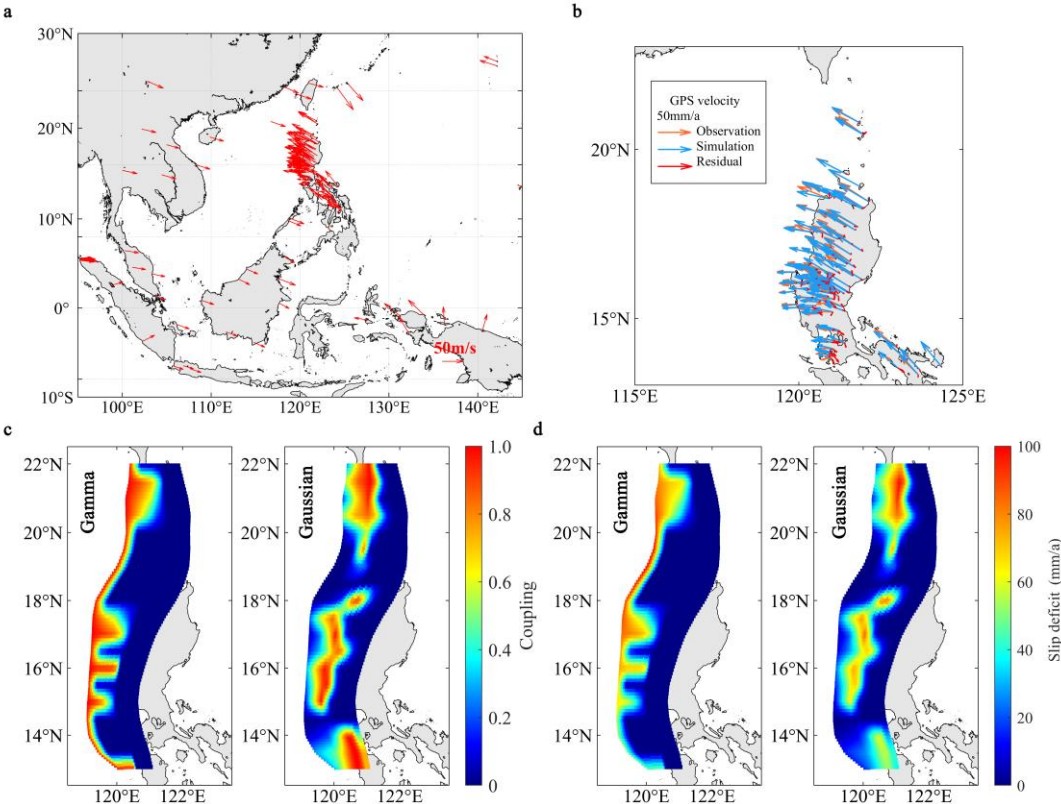

**Figure 1. Basic data and inversion results. (a) The GPS velocity data around the Manila subduction zone used for inversion. (b)**
**The residual between inversion results and observed GPS velocity. (c) The coupling/locking rate of the Manila subduction zone as the result of inversion. (d) The slip deficit rate of the Manila subduction zone as the result of inversion.**

When the shear modulus $\mu = 4 \times 10^{10}\,\text{N/m}^2$, the cumulative rate of seismic moments along the entire fault plane of the Manila Trench is $2.20 \times 10^{20}$ N·m/a in the Gaussian distribution and $1.63 \times 10^{20}$ N·m/a in the Gamma distribution. According to the historical earthquake database of the United States Geological Survey (USGS) from 1900 to 2022, the annual seismic moment release rate is estimated as $3.68 \times 10^{19}$ N·m/a in the last 123 years. Assuming that the future seismic moment accumulation rate and release rate remain unchanged, the actual seismic moment accumulation rate in the two models is $1.83 \times 10^{20}$ N·m/a and $1.26 \times 10^{20}$ N·m/a, respectively. By accumulating seismic moments at this rate, the maximum earthquake magnitudes that can be induced by accumulated seismic moments in 200 years are 9.0 and 8.9, and the maximum earthquake magnitudes in 500 years are 9.3 and 9.2.

However, many studies have shown that the Manila fault is unlikely to fully rupture in one earthquake event. The plate tearing at 17°N (Bautista et al., 2001), the drastic changes in dip and strike angles near 16 to 17°N (Yu et al., 2018), the sharp bend of Manila Trench between 14 and 14.5°N (Zhu et al., 2023) or the Huangyan (Scarborough) seamount near 15°N (Hsu et al., 2012; Zhu et al., 2023) may inhibit the propagation of rupture. Therefore, referring to the aforementioned studies, the Manila subduction zone is divided into two segments with a boundary at 14.5°N. The range of the north segment is 14.5 to 22°N and the south segment is 13 to 14.5°N.

In addition, the static cumulative seismic moment estimated by the locking may only be released by about 40~50%, which is caused by the non-uniformity of faults (Yang et al., 2019; Yao & Yang, 2022). Therefore, the release rate of 40% of static cumulative seismic moment is considered when evaluating the seismic potential. The seismic moment accumulation rate and maximum earthquake magnitudes for each segment for 200 and 500 years are estimated as shown in Table 1. The Manila subduction zone has had no record of large earthquake for over 400 years. The maximum earthquake magnitude in the Manila subduction zone is estimated to be $M_s$ 8.0 in 1619 (Bautista and Oike, 2000). This study takes the maximum magnitude of 8.9 for 500 years as an example for the tsunami hazard assessment. The impact of different magnitude limits corresponding to different recurrence periods on PTHA is discussed in the following section.

**Table 1. Seismic moment accumulation rate and maximum magnitude for each segment**

|  |  | Gaussian | Gamma |
|---|---|---|---|
| South segment (13°N ~14.5°N) | Seismic moment accumulation rate (N·m/a) | $3.08 \times 10^{19}$ | $0.34 \times 10^{19}$ |
|  | Maximum magnitude for 200 years | 8.2 | 7.6 |
|  | Maximum magnitude for 500 years | 8.5 | 7.9 |
| North segment (14.5°N~22°N) | Seismic moment accumulation rate (N·m/a) | $15.22 \times 10^{19}$ | $12.23 \times 10^{19}$ |
|  | Maximum magnitude for 200 years | 8.7 | 8.6 |

| Maximum magnitude for 500 years | 8.9 | 8.9 |
|---|---|---|

### 3.2 Spatial distribution of tsunami hazard in the South China Sea

Tsunamis generated by possible earthquake scenarios have been obtained by numerical model to show the spatial distribution of tsunami hazard. Considering the uncertainty of earthquake magnitude and epicenter, 20,073 possible combinations of magnitude and epicenter were adopted, when the maximum magnitude of the Manila subduction zone had been evaluated as 8.9. For each combination, 302 different heterogeneous slip distribution and 1 uniform slip distribution scenarios were considered. A total of 6,082,119 tsunami scenarios were simulated using the unit source method (Zhang & Niu, 2020). The observation points in the South China Sea were selected at an interval of 0.1°, with a total of 23,754 observation points. The tsunami wave height of each observation point with a return period of 1000 years can be obtained through the PTHA method (Fig. 2a).

The main high tsunami risk regions in the South China Sea include the southeast coast of China and the southern part of Taiwan Island, the western coast of Luzon Island in the Philippines, and some islands and reefs in the South China Sea include the Xisha Islands and the Dongsha Islands. The tsunami wave heights in these regions are generally greater than 4 m with a return period of 1000 years. Luzon Island faces a high tsunami risk due to its proximity to the source area. As the continental shelf outside the southeast coast of China becomes shallower, the tsunami waves gradually increase. The tsunami wave height on the east side of the southeast coast is significantly higher than that on the west side. The steep terrain of Taiwan Island and Xisha Islands also makes these areas face high tsunami risk. However, these islands only have a high tsunami risk on the side facing the source area.

Compared with previous study on the spatial distribution of tsunami hazard in the South China Sea (Ma et al., 2022), the spatial distribution pattern of tsunami hazard is similar, and the location of high-risk areas is the same. The spatial distribution map has important reference significance for hazard prevention and mitigation, as well as the design and construction of offshore engineering in the South China Sea. The tsunami wave height with a return period of 1000 years in the South China Sea obtained in this study is slightly smaller, and possible reasons include the selection of upper magnitude limit and fault segmentation, which will be discussed in detail in the following section.

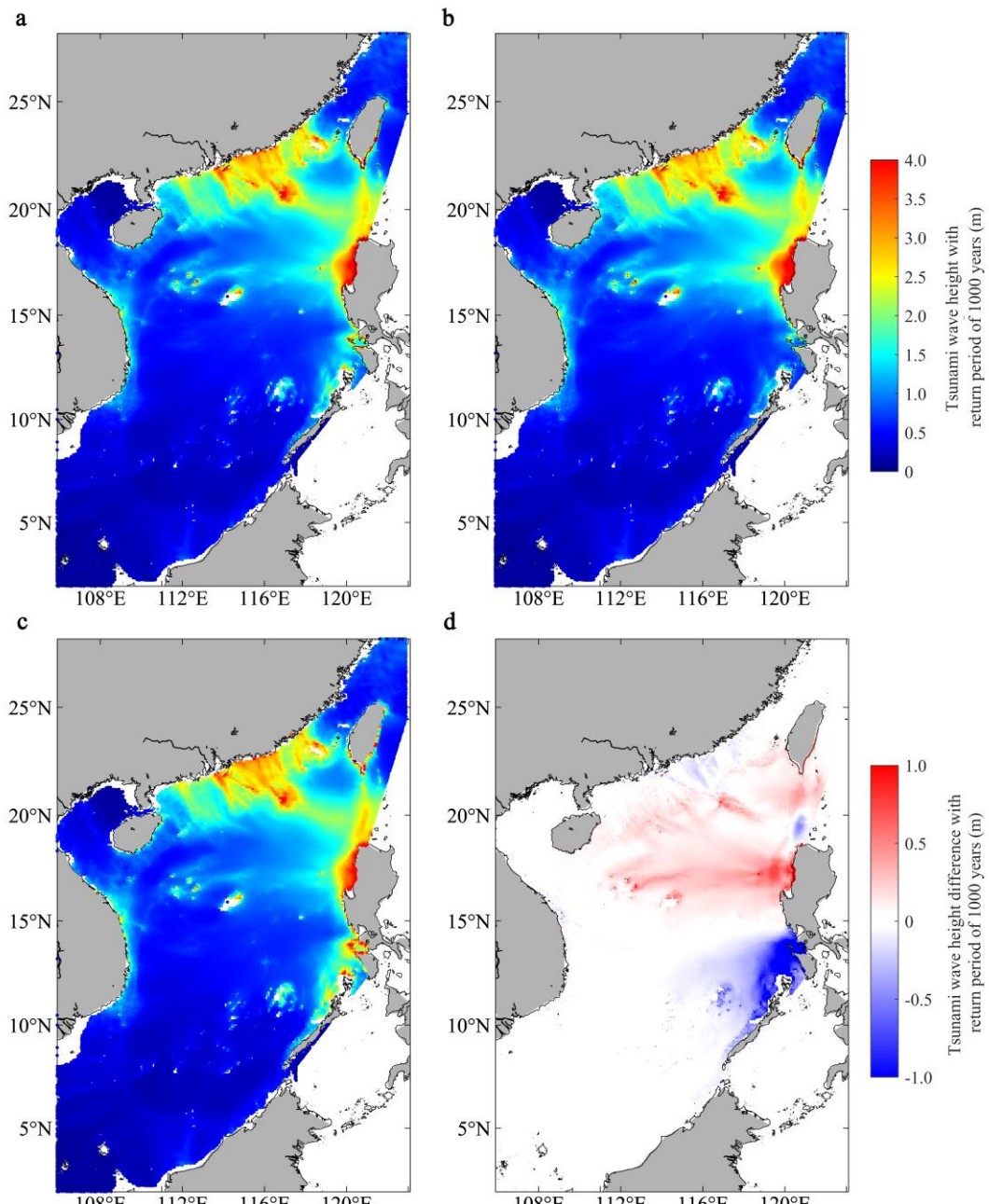

**Figure 2. Spatial distribution of tsunami hazard with a return period of 1000 years in the South China Sea. (a) Mean tsunami wave height with a return period of 1000 years. (b) Tsunami wave height with a return period of 1000 years in Gamma model. (c) Tsunami wave height with a return period of 1000 years in Gaussian model. (d) The difference of tsunami wave height with a return period of 1000 years between Gamma model and Gaussian model.**

## 3.3 The impact of source uncertainty on PTHA

The heterogeneous slip is the sum of stochastic slip and the deterministic slip scaled by the locking distribution. Considering the locking distribution as Gamma model or Gaussian model does not bring much difference (Fig. 2b and 2c). The main differences are concentrated on the west coast of the Philippines (Fig. 2d). In the Gamma model, slip tends to concentrate in 230 shallower areas, resulting in larger tsunami hazard. In the north segment with the same maximum magnitude, there are larger tsunami hazard in the Gamma model. However, in the south segment of the source area, the upper limit of the magnitude of the Gamma model is significantly lower than that of the Gaussian model. Therefore, in the Gaussian model, Palawan Island and Mindoro Island in the southwest of Luzon Island also have a high tsunami risk, while the Gamma model does not.

Whether it is stochastic slip or deterministic slip scaled by locking distribution, slip is more concentrated in a certain area, 235 making the slip in that area much larger than the average slip. Compared with the uniform slip model, the influence of slip heterogeneity on tsunami wave height are analyzed, taking observation points of Sanya and Hong Kong as examples (Fig. 3a and 3b).

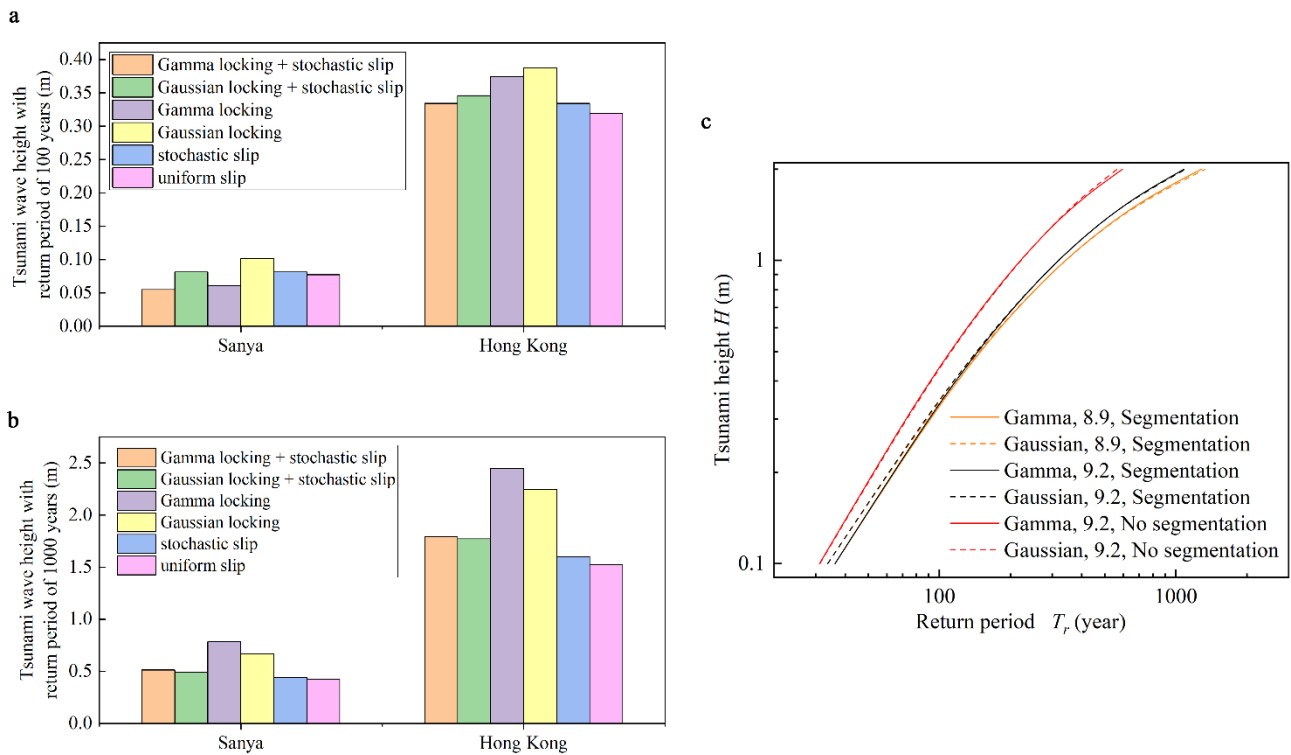

**Figure 3. The impact of source uncertainty on tsunami hazard assessment. (a) The tsunami wave height with a return period of** 240 **100 years of Sanya and Hong Kong in different heterogeneous slip scenarios. (b) The tsunami wave height with a return period of 1000 years of Sanya and Hong Kong in different heterogeneous slip scenarios. (c) The tsunami hazard curves of Hong Kong in scenarios with different upper limit of magnitude and segmentation.**

In general, the tsunami wave height will increase by an average of 5% when considering the slip heterogeneous. This value may be underestimated because in the Gamma locking model, the upper magnitude limit of the south segment was reduced, resulting in a decrease in the tsunami hazard level. This is also why the tsunami hazard is relatively low in the results of considering Gamma locking distribution with the return period of 100 years. When the return period is 100 years, the scenarios with the largest tsunami wave heights are those that only the Gaussian locking distribution is considered without the stochastic slip, with an increase of 21% at Hong Kong compared to uniform slip scenarios. When the return period is 1000 years, the scenarios with the largest tsunami wave heights are those that only the Gamma locking distribution is considered, with an increase of 60% at Hong Kong compared to uniform slip scenarios. In Gamma locking distribution, slip is assumed mainly concentrated in shallow areas of the subduction zone, which will increase the tsunami hazard when the range of earthquake rupture includes shallow areas of the subduction zone. For small magnitude earthquakes, only a small number of potential earthquakes are affected due to the small range of earthquake rupture. For earthquakes with large magnitude, most potential earthquakes are affected due to the large rupture range. When the recurrence period is short, the tsunami hazard level is mainly affected by small magnitude earthquakes, and the Gamma locking distribution produces a lower level of tsunami hazard due to the lower upper limit of magnitude in the south segment. When the recurrence period is long, the tsunami hazard level is affected by earthquakes with large magnitude, and the Gamma locking distribution that considers fault slip occurring in the shallower area of the subduction zone, may results in large tsunami waves. Meanwhile, the impacts of heterogeneous slip do not have a consistent trend at different locations. This is related to the propagation characteristics of tsunami waves. Different heterogeneous slip distributions cause the initial water field of the tsunami to concentrate in different region, thereby affecting the propagation of the tsunami wave. At specific locations, the amplitude of tsunami waves generated by highly concentrated slip may not necessarily be large. But by comparing the tsunami hazard in the overall sea area, similar patterns can still be obtained.

Whether the Gamma locking distribution or the Gaussian locking distribution is used for heterogeneous slip, slip always tends to concentrate in the shallow part of the fault, which leads to large tsunami hazard. Compared to uniform slip with the same upper limit of magnitude of north and south segments, considering only stochastic slip will lead to an average increase of 8% in tsunami hazard. Considering only the Gaussian locking distribution will lead to an average increase of 13%. But considering both will lead to an increase of 6%. This may be because the high slip regions are not always the same, so the peak of slip is weakened after the two distributions are stacked and scaled to meet the same seismic moment.

Although the increase in tsunami hazard caused by stochastic slip is not as significant as the locking distribution, stochastic slip has strong randomness and may cause significant differences in different regions. The maximum increase of stochastic slip at a single target point is 36%, which is greater than 25% considering only the locking distribution when the return period is 100 years. Therefore, for a single observation point, the influence of slip heterogeneity cannot be ignored.

There is still significant uncertainty in the upper limit of magnitude and segmentation methods. Although the magnitude formed by the seismic moment accumulation of 500 years is selected as the maximum possible magnitude in this study, the selection is subjective, and there is still large uncertainty in the locking distribution itself. Therefore, there is also large

uncertainty in the estimated maximum magnitude. At the same time, there are different suggestions for fault segmentation methods in different studies. The upper limit of magnitude and the segmentation affect the size and possible location of the largest earthquake. Taking Hong Kong observation point along the southeast coast of China as example, hazard curves were made for different scenarios (Fig. 3b).

Keep the upper limit of magnitude for the south segment unchanged and increase the upper limit of magnitude for the north segment from 8.9 to 9.2, which is the magnitude used in previous study (Zhang & Niu, 2020). On average, by raising the upper limit of magnitude, the tsunami wave height with a return period of 100 years increases by 4.01%, and 8.23% for tsunami wave height with a return period of 1000 years. The impact of increasing the upper limit of magnitude on tsunami wave height is not significant. Possible reasons include: only changing the upper limit of the north section; the probability of large magnitude earthquakes is very low, which has relatively small impact on overall PTHA. It should be noted that the increase of upper limit of magnitude has higher impact on tsunami wave height of longer return period. This is because the tsunami wave heights with longer return period are usually caused by the earthquake with larger magnitude.

Cancel the north-south segmentation and limit the magnitude of the Manila subduction zone to 9.2, compared to the scenarios where the upper limit of magnitude of the north segment is 9.2. On average, by canceling the north-south segmentation, the tsunami wave height with a return period of 100 years increases by 45.60%, and 87.01% for tsunami wave height with a return period of 1000 years. The impact of segmentation methods on tsunami hazard is significant. After segmentation, due to the different maximum magnitudes on different segments, the probability of earthquakes with the maximum magnitude changes significantly, which in turn has a huge impact on the level of tsunami hazard. Therefore, it is necessary to carefully consider the segmentation in future research.

**4 Discussion and Conclusions**

The study provides the spatial distribution of tsunami hazard in the South China Sea through PTHA. By introducing the geodetic locking model, possible solutions have been proposed for two key issues in PTHA: how large the upper limit of the magnitude is and how the slip distribution should be constrained. By inverting the GPS horizontal velocity data, the locking distribution and slip deficit distribution of the Manila subduction zone are obtained. The cumulative rate of seismic moment is calculated according to the distribution of slip deficit, and then the maximum possible magnitude is estimated. At the same time, using the distribution of slip deficit as a constraint on stochastic slip increases the likelihood of larger slip in higher locking area. In addition, the impact of uncertainty sources on PTHA was explored, including the slip heterogeneity, upper limit of magnitude and segmentation.

Existing studies rarely consider the influence of geodetic locking on the distribution of slip. Geodetic locking reflects the location of high slip regions on the earthquake rupture. In traditional stochastic slip models, the high slip regions are usually randomly generated on the fault plane, which is independent of the stress accumulation on the fault plane. In the improved heterogeneous slip model, the high locking region has more margin to produce larger slip, which means a higher probability

of producing larger slip. Research has found that the locking distribution significantly concentrates the fault slip, and its
amplification effect on tsunami wave height is more significant than stochastic slip. This emphasizes the importance of considering the locking distribution in future PTHA research. The geodetic locking results used in this study still have great uncertainty, which is a limitation of this study. In the future, we should deploy more observation stations near deep-sea trenches to improve the accuracy of the locking inversion.

In addition, the study found that segmentation has a significant impact on tsunami hazard assessment, which has rarely been discussed in previous studies. The geological structure of the Manila subduction zone has significant north-south differences, with the central seamount region as the boundary. However, there is significant uncertainty about the specific subduction location of the seamount, and there is uncertainty about whether it can block the propagation of earthquake rupture along the trench. Therefore, segmentation affects the seismogenic mechanism of the Manila subduction zone and directly determines the magnitude of the largest earthquake it can produce. Therefore, the accuracy of segmentation determines the accuracy of tsunami hazard assessment. In the future, the work of seismicity and tsunami hazard assessment of the Manila subduction zone should be combined with seismology and geology, focusing on detecting the fine structure of the shallow part of the subduction zone, providing scientific data constraints for studying the seismogenic mechanism and segmentation characteristics of the fault.

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

**Acknowledgments**

The author would like to acknowledge the support by the National Natural Science Foundation of China under the grant No. 51779125, and the support by State Key Laboratory of Hydroscience and Engineering under grant No. 2022-KY-05 and Tsinghua University Initiative Scientific Research Program under grant No. 20233080025.

**Code/Data availability**

The data and software used in this paper are open-source. The GPS velocity data used for fault locking inversion in the study are available via https://doi.org/10.1002/2014GC005407. The TDEFNODE (Version 2022.11.03) used for fault locking inversion are available via https://robmccaffrey.github.io/TDEFNODE/. The FVCOM (Version 3.2.1) used for tsunami simulation are available via https://www.fvcom.org/.

**Author contribution**

X. Niu designed the research. G. Zhao prepared the simulation codes and wrote the manuscript. X. Niu and G. Zhao contributed to data interpretation and provided discussions to improve the quality of the paper.

**Competing interests**

The authors declare no competing interests.