# Peer review of "Tsunami Hazard Assessment in the South China Sea Based on Geodetic Locking of the Manila Subduction Zone"

_Natural Hazards and Earth System Sciences, 2023_

## Referee Comment (RC1)

Evaluation:

This is an important study which estimate the spatial distribution of tsunami hazard in the South China Sea based on Geodetic locking of the Manila subduction zone. The study should be published because the influence of geodetic locking on the distribution of slip is rarely considered on tsunami hazard assessment in the South China Sea region among the current researches. It can help to understand the influence of uncertainties of the seismic source on tsunami hazard assessment. The article is well organized and well written. The present manuscript only needs minor revision for Natural Hazards and Earth System Science publication with the following comments.

Comments:
1. In general, the English of the text is good, but could be further improved. If you can, please ask a native speaker to polish the text to improve its readability.

2. At present, the abstract part does not give a good overview of the innovative points of the article. Please further summarize it.

3. Please check terminology consistency throughout the text. Such as "the maximum possible magnitude" and "the possible maximum magnitude", as we all know, they represent different meanings.

4. Line 258: "in the current researchs" should be "in the current researches"; Please check out.

5. In the introduction part, it will be good that the quantitative tsunami hazard assessment results from other researchers should be addressed and cited.

6. The impact of source uncertainty on tsunami hazard assessment in Figure 3a, why does the authors use the 100-year return cycle as an example instead of 1000 years? In addition, we can find that the impacts did not have a consistent trend at different locations with the same heterogeneous slip scenarios. Please add possible reasons for the results.

7. It might be good to provide brief discussions on the limitation of the present method of based on geodetic locking especially for tsunami hazard assessment.

---

## Author Comment (AC1)

**Authors' response to the reviewer's comments**

**Title:** Tsunami Hazard Assessment in the South China Sea Based on Geodetic Locking of the Manila Subduction Zone

We are very grateful to the reviewer for the comments and suggestions to improve the manuscript. Following the reviewer's suggestion, the manuscript has been revised accordingly.

This is an important study which estimate the spatial distribution of tsunami hazard in the South China Sea based on Geodetic locking of the Manila subduction zone. The study should be published because the influence of geodetic locking on the distribution of slip is rarely considered on tsunami hazard assessment in the South China Sea region among the current researches. It can help to understand the influence of uncertainties of the seismic source on tsunami hazard assessment. The article is well organized and well written. The present manuscript only needs minor revision for Natural Hazards and Earth System Science publication with the following comments.

Response: Thank you very much for your review work and valuable suggestions. These will also be of great help to our future work.

**Comments:**

1. In general, the English of the text is good, but could be further improved. If you can, please ask a native speaker to polish the text to improve its readability.

Response: The text has been further polished.

2. At present, the abstract part does not give a good overview of the innovative points of the article. Please further summarize it.

Response: The innovation points of the article include providing a dataset of tsunami hazard in the South China Sea and considering the locking distribution in the analysis which make the slip distribution and assessment results more realistic. The abstract has been revised to include the innovative points regarding the impact of locking distribution on tsunami hazard assessment, as follows:

Moreover, the assessment results involving the effect of locking distribution should be more realistic, and show a larger tsunami height than only considering the stochastic slip in most areas, which prompt the coastal management agencies to enhance the tsunami prevention awareness.

3. Please check terminology consistency throughout the text. Such as "the maximum possible magnitude" and "the possible maximum magnitude", as we all know, they represent different meanings.

Response: The terminology consistency has been checked and the "maximum possible magnitude" has been uniformly adopted.

4. Line 258: "in the current researchs" should be "in the current researches"; Please

check out.

Response: "In the current researchs, the influence of geodetic locking on the distribution of slip is rarely considered" has been changed to "Existing studies rarely consider the influence of geodetic locking on the distribution of slip".

5. In the introduction part, it will be good that the quantitative tsunami hazard assessment results from other researchers should be addressed and cited.

Response: Some researches about the probabilistic tsunami hazard assessment and their results at Hong Kong has been addressed and cited in the introduction part. For example, Li et al. (2016) studied the impact of uniform and heterogeneous slip distribution on the tsunami hazard assessment and the tsunami wave height with 1000-year return period of Hong Kong is about 2.0 m. Li et al. (2017) studied the role of upper magnitude limits in probabilistic tsunami hazard assessment and the tsunami hazard of Hong Kong at return period of 1000 years are about 0.5~3.5 m. Sepúlveda et al. (2019) conducted probabilistic tsunami hazard assessment focusing on the sensitivity to earthquake recurrence relationships, the maximum tsunami amplitude of 0.18 m is exceeded in Hong Kong with a mean return period of 100 years. Liu et al. (2021) considered the local and regional tsunami sources and the tsunami wave height of Hong Kong is 0.32 m for 475-year return period and 0.50 m for 975-year return period. Yuan et al. (2021) considered the tsunami source from both the South China Sea and the Northwest Pacific Ocean and the maximum wave amplitude of Hong Kong is about 2.5 m for 2000-year return period and 1.5m for 500-year return period.

6. The impact of source uncertainty on tsunami hazard assessment in Figure 3a, why does the authors use the 100-year return cycle as an example instead of 1000 years? In addition, we can find that the impacts did not have a consistent trend at different locations with the same heterogeneous slip scenarios. Please add possible reasons for the results.

Response: The map of tsunami hazard of 100-year return period and 1000-year return period were both obtained, but the manuscript only showed the results of 1000-year return period. In the comparison of Figure 3, the results of 1000-year return period are added. And an analysis is conducted on the differences in results between the tsunami wave heights of 100-year return period and 1000-year return period. At the same time, an analysis of the differences in the patterns of change at different locations is added, as follows:

When the return period is 100 years, the scenarios with the largest tsunami wave heights are those that only the Gaussian locking distribution is considered without the stochastic slip, with an increase of 21% at Hong Kong compared to uniform slip scenarios. When the return period is 1000 years, the scenarios with the largest tsunami wave heights are those that only the Gamma locking distribution is considered, with an increase of 60% at Hong Kong compared to uniform slip scenarios. In Gamma locking distribution, slip is assumed mainly concentrated in shallow areas of the subduction zone, which will increase the tsunami hazard when the range of earthquake rupture includes shallow areas of the subduction zone. For small magnitude earthquakes, only

a small number of potential earthquakes are affected due to the small range of earthquake rupture. For earthquakes with large magnitude, most potential earthquakes are affected due to the large rupture range. When the recurrence period is short, the tsunami hazard level is mainly affected by small magnitude earthquakes, and the Gamma locking distribution produces a lower level of tsunami hazard due to the lower upper limit of magnitude in the south segment. When the recurrence period is long, the tsunami hazard level is affected by earthquakes with large magnitude, and the Gamma locking distribution that considers fault slip occurring in the shallower area of the subduction zone, may results in large tsunami waves. Meanwhile, the impacts of heterogeneous slip do not have a consistent trend at different locations. This is related to the propagation characteristics of tsunami waves. Different heterogeneous slip distributions cause the initial water field of the tsunami to concentrate in different region, thereby affecting the propagation of the tsunami wave. At specific locations, the amplitude of tsunami waves generated by highly concentrated slip may not necessarily be large. But by comparing the tsunami hazard in the overall sea area, similar patterns can still be obtained.

7. It might be good to provide brief discussions on the limitation of the present method of based on geodetic locking especially for tsunami hazard assessment.
Response: The limitation of the tsunami hazard assessment based on geodetic locking is provided in the Discussion and Conclusions. There is still great uncertainty in the geodetic locking and fault segmentation results in this study due to the limited understanding of locking and segmentation at present, resulting in limitations of the present method.

---

## Author Comment (AC2)

**Authors' response to the reviewer's comments**

**Title:** Tsunami Hazard Assessment in the South China Sea Based on Geodetic Locking of the Manila Subduction Zone

We are very grateful to the reviewer for the comments and suggestions to improve the manuscript. Following the reviewer's suggestion, the manuscript has been revised accordingly.

The manuscript "Tsunami Hazard Assessment in the South China Sea Based on Geodetic Locking of the Manila Subduction Zone" presents a new tsunami hazard assessment for the South China Sea due to earthquakes triggered in the Manila Subduction Zone. The new assessment incorporates geodetic information that is used to increase the likelihood of hosting larger slip within highly coupled regions and to determine an upper limit for the maximum earthquake moment magnitude. The study provides a significant advance to the efforts assessing tsunami hazards in the South China Sea. Here, I provide some comments to further improve the paper.

Response: Thank you very much for your review work and valuable suggestions. These will also be of great help to our future work.

**Major Comments:**

1. The paper needs a deeper analysis of previous geodetic studies in the Manila Subduction Zone. For example, I would include a deeper analysis of previous GNSS data used by Hsu et al. (Hsu, 2016, 2012) in the introduction. That study served as reference for several PTHA studies in the past (to define earthquake magnitude recurrences). It is also important to compare differences in the estimated coupling ratios of past studies to identify improvements in your new inversion.

Response: The analysis of previous geodetic studies and estimated locking ratios of past studies has been provided in the Introduction, as follow:

So far, there already are some studies on locking inversion in the Manila subduction zone. Galgana et al. (2007) showed that the locking degree of the Manila subduction zone is very low, with a locking coefficient of 0.01. Hsu et al. (2012) suggested that the Manila subduction zone is partially locked between 14.5-17.0°N, with an average locking coefficient of 0.4. Hsu et al. (2016) estimated that the locking coefficient of the Manila Trench at 15.0-19.0°N is 0.34~0.48. Those works are good references for the present study, but further analysis is still needed for deeply integrating the effect of locking distribution into PTHA.

2. The locking model is a very important input in the new tsunami assessment. Because of this, I recommend to add more information about the inversion method. For example, you need to provide information on how the Gaussian and Gamma distribution enter in the inversion method. Also, you shall need to provide some measure of the inversion constraints. For example, your slip seems to concentrate in shallow regions. Is this a bias due to the GPS station locations? This information will be very important for future

efforts to improve the geodetic network or understand tsunami hazard uncertainties.

Response: The information on how the Gaussian and Gamma distribution enters in the inversion method and measure of the inversion results are provided. Generally, the locking distribution along the dip profile is assumed to have similarity, and parameterized functions are used as a primary guess of locking distribution. The Gaussian function and the Gamma function are widely used in the locking inversion. The Gaussian type refers to the distribution of locking coefficients along the dip profile as a Gaussian function; Gamma type refers to the exponential distribution of locking coefficients along the dip profile. The goal of inversion is to find the optimal parameters of the assumed function that minimizes the chi-square value between the observed data and the model data. In Gamma distribution, it is assumed that the locking coefficient is maximum at the shallowest part of the subduction zone, so the locking distribution and slip deficit distribution are concentrated in the shallow regions.

3. Due to computational limitations, the tsunami modeling of the study is based on linear superposition of unit sources. This approach has to assume linear tsunami waves. Though, tsunami waves are very non-linear in shallow waters. Because of this, the study would not be able to determine tsunami heights in coastal regions. If the linear superposition at the coast is used, I'm afraid the results will be very different from an approach using non-linear tsunami models. The authors, therefore, may be only allowed to determine wave heights in relatively deep waters (at some distance from the coast) and for waves that have not propagated through shallow waters before. This is an important limitation which can be only overcome by running non-linear models (where the unit source superposition is not valid). Furthermore, bottom friction will contribute with energy dissipation. This linear superposition limitation is discussed in several papers (Williamson et al., 2020; Sepulveda et al., 2019; Li eta al., 2017). An easy way to overcome this difficulty would be to analyze tsunami heights far from the coast. This would be easy as the authors already have the model results in all the domain. Finally, the innovative idea of using linear superposition (in the past) was designed before Zhang and Niu (2020). For example, Li et al. 2016. Please indicate if there is something different in the most recent cited paper or change the citation.

Response: You are right that the tsunami modeling of this study would not be able to determine tsunami heights in coastal regions. The water depth of target points in this study is all greater than 100 m. Earlier papers on linear superposition techniques have been cited. This study used the same unit source database with study of Zhang and Niu (2020), so this paper is still cited. The description related to linear superposition limitation has been supplemented in Section 2.3, as follows:

It should be noted that due to the limitation of the superposition method, this study would not determine tsunami heights in coastal regions and the water depth of target points is all greater than 100 m. Tsunami waves propagating in shallow water will show complex non-linear behaviours, which can be simulated using non-linear models. The tsunami height in shallow water can be obtained approximately from the tsunami wave height at offshore point such as 100 m depth through multiplying the nearshore amplification factors (Glimsdal et al., 2019; Gao et al., 2022). Generally, the tsunami

dataset in this study can be adopted as the boundary condition for detailed nearshore hazard analysis.

4. The abstract does not mention the very important innovations of the study. It rather focuses on ideas that are well-known from the past. For example, lines 12-14 compares the new slip model with the old uniform-slip model. Rather than this, I would tell how relevant is to include the locking distribution in the new slip model, compared to a slip model that only uses stochastic slip.
Response: The abstract has been revised to include the innovative points regarding the impact of locking distribution on tsunami hazard assessment, as follows:
    Moreover, the assessment results involving the effect of locking distribution should be more realistic, and show a larger tsunami height than only considering the stochastic slip in most areas, which prompt the coastal management agencies to enhance the tsunami prevention awareness.

5. I could not see the details of the PTHA. What are the recurrences for every magnitude (i.e., p_i in Equation 1)? Are you able to determine new a and b value to create a Gutenberg-Richter Law? Please clarify as this is a very important input for future studies.
Response: The detail of PTHA has been provided. $p_i$ is estimated statistically using the historical earthquake data. The historical earthquake data from 1900 to 2022 of the United States Geological Survey (USGS) is used to calculate the coefficients in the Gutenberg-Richter relationship.

6. Line 121-125. Here I got confused. First, it says that the Okada (1985) model is used to get the tsunami initial condition unit (which assumes that water elevation mimics co-seismic deformation). Though, in line 124 you say "same Gaussian distribution initial water level is set on each point source". Please clarify.
Response: Additional description has been provided. Unit sources have the same initial water level field of Gaussian distribution. The initial water level distribution of unit sources will be stacked according to certain proportion coefficients to obtain the initial water level field of tsunami event calculated by the Okada model.

7. The introduction contains some ideas which may be true some years ago but not today: Line 44: "traditional tsunami research often assumes that earthquake rupture is uniform". I think this is not true anymore. I rarely see tsunami assessments with uniform slip, even in the engineering industry. Line 48: "…unit source or sub-fault methods are usually used to convert tsunami simulation into linear superposition...". I also think, the linear superposition is not used so often. Especially because tsunami hazards are commonly evaluated close to the coast.
Response: The relevant sentences have been revised. For example, some traditional tsunami research assumed that earthquake rupture was uniform. And in some PTHA work, unit source or sub-fault methods are used to convert tsunami simulation into linear superposition of unit sources based on the linear characteristics of tsunami waves

in deep water, thereby reducing the computational complexity.

8. To make the manuscript reproducible, the results need to be provided in accessible files. Are the coupling and locking rates included in a text file or repository in "Code/Data Availability". These are essential to reproduce results.
Response: They are not in Code/Data Availability for now. Readers can obtain locking data from authors via email.

**Minor Comments:**
1. Line 66: I would replace "increase the probability" by "correct the probability"
Response: The text has been revised.

2. Line 72-73: I would remove the sentence "TDEFNODE is an inversion program developed by Professor McCaffrey of Portland State University in the United States" and rather talk about the method and formulations (which are missing in the text).
Response: The text has been revised and the method of TDEFNODE has been supplemented.

3. Line 88: what constitutes 'great uncertainty" for the study? Please provide quantification or justification.
Response: If the velocity data uncertainty of a GPS station is greater than 3.3 mm/a, this data is thought to have great uncertainty and is eliminated.

4. Line 147: "…1.26x10^20 N*m/a, respectively."
Response: The text has been revised.

5. Line 153: maybe use "the aforementioned studies" instead of "the latest research results", to emphasize that you are talking about the papers you described in the previous sentence.
Response: The text has been revised.

6. Line 156: "locking may only be released" instead of "locking can only be released"
Response: The text has been revised.

7. Line 258: Sentence starting with "In the current researchs…" I would suggest "Existing studies rarely consider the…"
Response: The text has been revised.